# How Is Arachidonic Acid Metabolism in the Uterus Connected with the Immune Status of Red Deer Females (*Cervus elaphus* L.) in Different Reproductive Stages?

**DOI:** 10.3390/ijms24054771

**Published:** 2023-03-01

**Authors:** Angelika M. Kotlarczyk, Julia Kaczmarczyk, Olga Witkowska-Piłaszewicz, Małgorzata Kotula-Balak, Anna J. Korzekwa

**Affiliations:** 1Department of Biodiversity Protection, Institute of Animal Reproduction and Food Research of Polish Academy of Sciences (IAR&FR PAS), Tuwima 10 Str., 10-748 Olsztyn, Poland; 2Department of Large Animal Diseases and Clinic, Institute of Veterinary Medicine, Warsaw University of Life Sciences, 02-787 Warsaw, Poland; 3Department of Animal Anatomy and Preclinical Sciences, University Centre of Veterinary Medicine JU-UA, University of Agriculture in Kraków, Mickiewicza 24/28, 30-059 Craków, Poland

**Keywords:** uterus, condition, lymphocytes, reproduction, red deer

## Abstract

Reproductive and condition parameters’ dependency on immune status in seasonally reproducing ruminants such as red deer have not been outlined to date. We determined T and B blood lymphocytes; the concentration of IgG, cAMP, haptoglobulin, and 6-keto-PGF1α in blood plasma; and the mRNA and protein expression of PG endoperoxide synthase 2, 5-lipoxygenase, PGE2 synthase (PGES), PGF2α synthase (PGFS), PGI2 synthase (PGIS), leukotriene (LT)A4 hydrolase, and LTC4 synthase (LTC4S) in the uterine endo- and myometrium, on the 4th (N = 7) and 13th (N = 8) days of the estrous cycle, in anestrus (N = 6) and pregnancy (N = 8) in hinds. An increase in CD4+ T regulatory lymphocyte percentage during the estrous cycle and anestrus compared with pregnancy was recorded; the opposite effect was observed for CD21+ B cells (*p* < 0.05). cAMP and haptoglobin concentration were elevated during the cycle, as was IgG on the fourth day of the cycle, whereas 6-keto-PGF1α concentration was the highest in pregnancy, and the nearest in anestrus similarly were LTC4S, PGES, PGFS, and PGIS protein expression in the endometrium (*p* < 0.05). We showed an interaction between the immune system activation and AA-metabolite production in the uterus throughout different reproductive stages. IgG, cAMP, haptoglobin, and 6-keto-PGF1α concentrations are valuable candidates for markers of reproductive status in hinds. The results help expand our knowledge of the mechanisms underlying seasonal reproduction in ruminants.

## 1. Introduction

The correlations between uterine function, condition, and immune status are yet to be explored. The immune status is shaped by exogenous and endogenous processes of exposure and the fighting of pathogens to achieve the clearance of infections [1]. Haptoglobin is a hemolysis marker, and its concentration is elevated in infections and inflammation [2], whereas the general immune condition is shaped by immunoglobulins and evaluated using the IgG concentration [3]. IgG is the most abundant antibody isotype, representing approximately 75% of the total immunoglobulins [4]. Therefore, the IgG level is an important diagnostic factor for evaluating an animal’s immunocompetence [5]. The next blood marker of inflammation selected for this study is prostacyclin, which is generated by the vascular wall and functions as a vasodilator and inhibitor of platelet aggregation by increasing the cyclic adenosine monophosphate (cAMP) level. This secondary messenger in the intracellular signaling pathways has effects on innate and adaptive immune cells, including T and B lymphocytes [6]. The nervous and endocrine systems exploit the cAMP pathway to modulate immune responses. The elevation of the intracellular cAMP concentration leads to the inhibition of proinflammatory cytokine production and macrophage phagocytosis [6]. As a circulating indicator of the AA vascular state of the organism, we measured a metabolite of PGI2—6-keto-PGF1α—in the blood plasma.

The percentage of lymphocytes in the peripheral blood fluctuates, and influences immunocompetence. T-helper (CD4+) cells activate humoral and cell-mediated immunity, while T-cytotoxic (CD8+) cells conduct immune surveillance in the peripheral tissues and remove pathogens [7]. Before pregnancy, in the human endometrium, T cells constitute 50–60% of all the lymphocytes. During early gestation, in decidual tissue, approximately 5–20% of the CD45+ positive lymphocytes are T cells, and this percentage increases to 40–80% during the development of pregnancy. The endometrial and peripheral leukocytes in the cattle and sheep endometrium decrease during early and mid pregnancy [8]. No lymphocytes or macrophages are found in the sheep and bovine caruncle by the middle of gestation [9,10,11]. In the endometrium, T cells are dominant and crucial for implantation [9]. However, the influence of the immune status on reproduction has not been outlined in seasonally reproducing ruminants to date.

Prostaglandins (PGs) are produced from arachidonic acid (AA) via the cyclooxygenase (COX) pathway and, as proinflammatory mediators, are known to be the main factors in many pathological conditions [12]. However, during physiological reproductive processes and pregnancy, they are closely related to physiological inflammatory responses [13]. Arachidonic acid is transformed to PGH2 by PGE and F synthases (PGES and PGFS), which then synthesize PGE2 and PGF2α, respectively [14]. Prostaglandin E2 is an important mediator of diverse functions during the estrous cycle and at the time of the establishment of pregnancy in cows. Moreover, Arosh et al. [15] pointed out that, in the uterus, PGE2 receptors (PGER2 and PGER4) are coupled to adenylate cyclase, generating cAMP, and PGE2 is involved in cAMP production. Atroshi et al. [16] highlighted a positive correlation between the levels of cAMP and PGs (PGF2α and PGE2) in the blood plasma of healthy cows. In cattle, together with increased concentration of PGF2α, there was an increase in the abundance of PTGFR in the endometrium during behavioral estrus [17].

Beside PGs, leukotrienes (LTs) are involved in the regulation of uterine functions in cows [18]. In the endometrium, LTs (LTB4 and LTC4) are produced and secreted during different phases of the estrous cycle, and these differences indicate a potential involvement of LTs in mechanisms related to proliferation or tissue remodeling in the endometrium [18]. Arachidonic acid is metabolized into LTs via the lipoxygenase (LO) pathway [19]. In addition to its crucial role in uterine functions, LTs also participate in inflammation [20]. Leukotriene B4 upregulates immune functions [21] and overrides the effects of progesterone (P4), but also plays a role in the stimulation of P4 and PGE2 secretions, whereas LTC4 elevates the secretion of PGF2α in the blood plasma in cattle [18,22]. Leukotrienes produced by proinflammatory cells (dendritic cells, neutrophils, macrophages, eosinophils, and mast cells) induce inflammatory responses by binding to and activating G-protein-coupled receptors in response to immune stimuli [21].

Based on knowledge concerning the action of AA metabolites in the ruminant uterus, we decided to explore the relationship between the immune status, immune condition, and uterine AA metabolism in the case of seasonally reproducing ruminants, such as red deer females. Knowledge of the correlation between the ruminant immune status and physiology of the uterus in red deer, which is photoperiod-dependent, is limited. We carried out a study in hinds aimed at determining during the estrous cycle, anestrus, and pregnancy: (i) T and B blood lymphocytes; (ii) the concentrations of circulating IgG, cAMP, haptoglobin, and a metabolite of PGI2 (6-keto-PGF1α) in the blood plasma; (iii) and the mRNA and protein expression of PG endoperoxide synthase 2 (PTGS2), 5-LO (5-lipoxygenase), PGES, PGFS, PG I synthase (PGIS), LTA4 hydrolase (LTA4H), and LTC4 synthase (LTC4S) in the uterine endometrium and myometrium.

## 2. Results

### 2.1. Determination of T and B Lymphocytes’ Activation Markers in the Blood

We demonstrated a lower percentage of CD21+ B cells in the blood in pregnancy compared with in the other reproductive stages (*p* < 0.001) (Figure 1A). There was no difference in the percentage of CD21+ B cells between the anestrus and the estrous cycle groups (*p* > 0.05). In addition, in pregnancy, there was a difference between the percentages of CD21+ B and CD21- cells (*p* = 0.0004 vs. *p* < 0.0001).

Moreover, a decrease in the percentage of CD4+CD8- helper T cells in the blood of pregnant red deer in comparison to anestrus (*p* = 0.01) on the 4th day of the estrous cycle (*p* = 0.02) and on the 13th day (*p* = 0.002) of the estrous cycle was observed (Figure 1B). By contrast, there was no difference in the percentage of CD4-CD8+ cytotoxic T cells between any of the reproductive stages (Figure 1B; *p* > 0.05). Additionally, in pregnancy, there was no difference in the percentage of CD4+CD8- and CD4-CD8+ T cells (*p* > 0.05). However, in anestrus and on the 4th and 13th days of the estrous cycle, the percentage of helper T cells CD4+CD8- was higher than that of cytotoxic T cells CD4-CD8+ (*p* = 0.001, *p* = 0.003, *p* < 0.0001, respectively).

In the pregnant females, there was a higher percentage of activated CD4+CD25+ T helpers in the blood than on the 4th (*p* = 0.01) and 13th day (*p* = 0.001) of the estrous cycle, but not in comparison to the anestrus (*p* = 0.056; Figure 1C).

In addition, the percentage of regulatory CD4+FoxP3+ T cells was higher in pregnancy than in anestrus, and on the 4th and 13th day of the estrous cycle (*p* = 0.001, *p* = 0.0007, *p* = 0.0001, respectively; Figure 1D).

### 2.2. IgG, cAMP, Haptoglobin, and 6-keto-PGF1α Concentration in Plasma

IgG concentration was elevated on the 4th day of the estrous cycle, and decreased on the 13th day of the cycle and in anestrus (*p* < 0.05; Figure 2A).

The level of cAMP was higher on the 4th and 13th days of the estrous cycle and in pregnancy compared with anestrus (*p* < 0.05; Figure 2B).

Haptoglobin concentration was the lowest on the 13th day of the estrous cycle, and increased in pregnancy and anestrus (*p* < 0.05; Figure 2C).

6-keto-PGF1α level was the highest in pregnancy, lower on the 4th and 13th day of the cycle, and the nearest in anestrus (*p* < 0.05; Figure 2D).

### 2.3. Arachidonic Acid Metabolites’ Abundances in Uterus

#### 2.3.1. mRNA Expression in Endometrium and Myometrium

In the endometrium, the mRNA expression for 5-LO was the highest on the 13th day of the estrous cycle, and differed from pregnancy and anestrus (*p* < 0.05; Figure 3A). LTC4S mRNA expression was the highest in pregnancy compared with the other reproductive stages, whereas the expression of LTA4H on the 4th day of the estrous cycle was the nearest compared with other reproductive stages (*p* < 0.05; Figure 3B,C). For PTGS2, PGES, PGFS, and PGIS, the highest mRNA expression was in pregnancy compared with in the other reproductive stages (*p* < 0.05; Figure 3D–G).

In the myometrium, no statistical differences were found for 5-LO mRNA expression compared with all the examined reproductive stages (*p* < 0.05; Figure 3A). LTA4H mRNA expression was the nearest on the 4th day of the estrous cycle compared with other phases (*p* < 0.05; Figure 3B). LTC4S mRNA expression was higher on the 13th day of the estrous cycle, and in pregnancy and anestrus compared with on the 4th day of the estrous cycle (*p* < 0.05; Figure 3C). The highest mRNA expression for PGES, PGFS, and PGIS was observed in pregnancy (*p* < 0.05; Figure 3E–G), and PGES mRNA expression was also higher in anestrus compared with the estrous cycle groups (Figure 3E; *p* < 0.05). For PTGS2 mRNA expression, no statistical differences between the examined reproductive stages were observed (*p* > 0.05; Figure 3D).

#### 2.3.2. Interaction between mRNA Expression in Endometrium and Myometrium

For 5-LO mRNA expression, an interaction on the 4th and 13th days of the estrous cycle and in anestrus between the endometrium and the myometrium was observed, and the mRNA expression was higher in the endometrium on the 4th and 13th days of the estrous cycle (*p* < 0.01; Figure 3A) and, during anestrus, was higher in the myometrium (*p* < 0.01; Figure 3A). For LTA4H and PGES, an interaction between the examined tissues occurred during anestrus, and the expression was the highest in the myometrium (*p* < 0.01 for LTA4H; Figure 3B and *p* < 0.001 for PGES; Figure 3E). For LTC4S and PGFS, an interaction in pregnancy was observed; however, for LTC4S, the expression was higher in the endometrium, and for PGFS, it was higher in the myometrium (*p* < 0.001; Figure 3C,F). In addition, for LTC4S, there was an interaction between tissues in anestrus, and the expression was higher in the endometrium than in the myometrium (*p* < 0.01; Figure 3C). An interaction for the mRNA expression for PGIS was observed in pregnancy, and the expression was higher in the endometrium (*p* < 0.01; Figure 3G).

#### 2.3.3. Protein Expression in Endometrium and Myometrium

In the endometrium, the protein expression of LTA4H, LTC4S, PGES, PGFS, and PGIS was the highest in pregnancy compared with the estrous cycle groups (4th and 13th days) and anestrus (*p* < 0.05; Figure 4B,C,E–G). For 5-LO, the protein expression was higher on the 4th and 13th days of the estrous cycle compared with other phases (*p* < 0.05; Figure 4A,D). No differences for PTGS2 were found throughout all the examined reproductive stages (*p* > 0.05; Figure 4D).

In the myometrium, the expression of 5-LO, LTC4S, and PTGS2 was similar in all the experimental reproductive stages (*p* > 0.05; Figure 4A,D), and no differences were found for any of them. LTA4H protein expression was higher in pregnancy and anestrus compared with the estrous cycle groups (*p* < 0.05; Figure 4B,C). The expression of PGES, PGFS, and PGIS was higher in pregnancy compared with the estrous cycle and anestrus, and differed from that in all the studied reproductive stages (*p* < 0.05; Figure 4E–G).

#### 2.3.4. Interaction between Protein Expression in Endometrium and Myometrium

An interaction for protein expression occurred for 5-LO and LTC4S, and the expression was higher in the endometrium on the 4th and 13th days of the estrous cycle for 5-LO (*p* < 0.1; Figure 4A) and in pregnancy for LTC4S (*p* < 0.001; Figure 4C).

## 3. Discussion

We determined, for the first time, the profile of the main enzymes connected with AA metabolism in both the red deer uterine endometrium and myometrium, and demonstrated that the production of PGs and LTs was correlated with the peripheral content of immune parameters in a manner dependent on the reproductive stage. Our study confirms that the immunological blood cellular response in red deer is similar to that in other mammals, and suggests that IgG, cAMP, haptoglobin, and 6-keto-PGF1α could serve as immune markers suitable for the evaluation of uterine functions in different reproductive phases. The relevance of changes in uterine immune function to the reproductive and immune status has not yet been fully established in seasonally reproducing females, such as red deer, and may have potential in the veterinary diagnostic of cervids and other seasonally reproducing ruminants.

### 3.1. Changes in Immunophenotype of Blood Lymphocytes

Immunological tolerance is critical for preventing the rejection of the fetus by the mother′s immune system during pregnancy. Shortly after the trophoblasts’ implantation, the number of maternal lymphocytes in the blood and endometrium change [23]. In humans, according to Lissauer et al. [24], during pregnancy, the number of the CD8+ cytotoxic T cells should be stable. This subpopulation has a negative influence on pregnancy because an increased number of these cells leads to miscarriages in humans and animals [25]. Similarly, in our study, there was also no difference in the percentage of CD8+ cytotoxic T cells between the pregnant and non-pregnant hinds.

Regulatory T cells CD4+ (Tregs) are essential for the maintenance of immune homeostasis [25]. In our study, there was a decrease in CD4+ cells in the blood of pregnant females in comparison to the estrus and anestrus group. This may be related to their transition to the endometrium, as has been reported in other species during pregnancy [25]. CD4+CD25+ Tregs represent a distinct lineage of naturally anergic and suppressive cells [26]. In our study, the percentage of this subpopulation in the blood of the pregnant group was increased, which confirms the establishment of immunological tolerance in the whole organism, allowing for the growth of a semi-allogeneic fetus in the uterus. The percentage of the suppressive Tregs is CD4+FoxP3+ increased in pregnant hinds’ blood as well. Tregs mediate suppressive activity mainly via reducing the capacity of dendritic cells (DCs) to present antigens by the secretion of anti-inflammatory cytokines such as IL-10 and TGF-β [27]. It was documented that these cells are potent mediators of self-tolerance and essential for the suppression of the immune responses triggered during pregnancy [28].

In addition, the percentage of CD21+ B cells was reduced in pregnant hinds. The suppression of B cell lymphopoiesis and B cell lymphopenia (responsible for immunoglobulin production) was documented in pregnant females [29], and explained as a mechanism tending to reduce the occurrence of autoreactive B cells that may recognize fetal structures and thereby cause pregnancy failures. The reduction in CD21+ B cells may also affect antibody production during pregnancy, which was also confirmed in our study.

### 3.2. Blood Markers of Condition Dependent on Reproductive Stage in Hinds

We observed the highest level of IgG at the beginning of the estrous cycle in hinds. Our explanation of such a result is the fact that in hinds, which, during the reproductive season, have only some ruts and, simultaneously, such a high IgG level strengthens the body and shows that the female is ready for fertilization. cAMP is regarded as a proquiescent factor that suppresses uterine contractions during pregnancy [30]. According to our results, the cAMP level in pregnancy was also lower than that during the estrous cycle. The nearest concentration of this secondary messenger was noticed in anestrus, where intensive enzymatic metabolism may not be needed in the females. Haptoglobin is mostly produced by the liver [2], but the uterine tubes and endometrium also produce haptoglobin in the preovulatory phase in cattle [31,32], and the secretion of haptoglobin is regulated by ovarian steroid hormones [31]. In the present study, the elevation of haptoglobin level may have occurred due to higher production from the reproductive tract, especially around ovulation. The pattern of 6-keto-PGF1α concentrations that we obtained indicates that this metabolite’s concentration in the blood is a convenient marker of local uterine blood flow in hinds during the reproductive season, in pregnancy and in anestrus.

It is not possible to conclude which of the markers we tested is the best for monitoring the reproductive status in hinds. However, based on the results obtained, it is known which of them shows a decrease or increase in a given reproductive period, e.g., the estrous cycle, pregnancy, or anestrus, and correlate the results with the expression of enzymes involved in AA metabolism in the uterus.

### 3.3. Expression Profile of Uterine Enzymes Involved in AA Metabolism in Red Deer Females

The PTGS2 mRNA expression was upregulated in the ovine endometrium on the 10–12th days [33] and 12–15th days [34] of the estrous cycle. Charpigny et al. [34] showed that PTGS2 protein was transiently expressed in the ovine endometrium between days 12 and 15 of the estrous cycle. In cattle, PTGS2 mRNA and protein were expressed at low and high levels on days 1–12 and 13–21 of the estrous cycle, respectively [35]. Our results show almost no differences in mRNA and protein expression between the studied reproductive seasons except for PTGS2 mRNA in the endometrium in pregnancy, where the expression was the highest, which highlights this enzyme as an inducible form in the hind uterus.

In the bovine endometrium, PGFS and PGES are highly expressed during the mid and late luteal phases of the estrous cycle [15]. Wocławek-Potocka et al. [36] showed higher PGES mRNA expression during pregnancy in stromal uterine cells compared with nonpregnant cows. However, for PGFS, higher expression was reported on days 16–18 of the estrous cycle in epithelial cells. The intuitive reason for the inconsistency between our and others’ results is that the synthesis of PGF2α is dependent not only on the conversion of PGH2 by the PGF synthases but also on the expression of other mediators, such as the cyclooxygenases (PTGS1 and PTGS2), for the conversion of AA into PGH2. In this regard, the production of PGH2 by the PTGS2 is considered the rate-limiting step of PG biosynthesis in the uterus [37]. The myometrium makes a considerable amount of PGI2, which is a uterine vascular smooth muscle relaxing agent. Rupnow et al. [38] showed increased myometrial PGIS protein expression during pregnancy in ewes. We obtained similar results, where PGIS mRNA and protein expression in both the endo- and myometrium were the highest in pregnancy.

We observed that 5-LO mRNA and protein expression were low in the first trimester of pregnancy and anestrus compared with the estrous cycle in hinds. Brown et al. [39] described higher 5-LO mRNA expression in the human decidua in the first trimester of pregnancy. However, Jian et al. [40] showed opposite results, where 5-LO mRNA expression in the human decidua was significantly higher during the third trimester of pregnancy versus early pregnancy. Our previous results showed that, in the bovine endometrium, 5-LO mRNA and protein expression were the highest in the early luteal stage of the estrous cycle [18]. In this study, we did not observe changes in 5-LO expression during the estrous cycle in hinds. Additionally, we previously showed the highest protein expression on days 2–4 of the estrous cycle for LTAH and on days 16–18 for LTC4S in the bovine endometrium. In the present study, we observed opposite results, where the highest mRNA and protein expression was on the 4th day of the cycle for LTA4H. The changes in the expression of LTC4S in the red deer uterus that we observed in the examined period were not dependent on the uterine layer. According to Molin et al. [41], the 5-LO pathway is involved in neovascularization and endothelial development, which suggests the participation of LTs in the regulation of uterine blood flow. Indeed, in the case of LTA4H expression, the increase at the beginning of the estrous cycle in both the endo- and myometrium may explain such LTB4 participation in vessel formation in hinds.

## 4. Material and Methods

### 4.1. Sample Collection

Twenty-nine (N = 29; 3–4-year-old) hinds were evaluated, and the age was confirmed according to Kotlarczyk et al. [42]. All individuals were healthy, which was confirmed using the hematological parameters analyses for hemoglobin, red blood cell count, packed cell volume, mean corpuscular volume, mean corpuscular hemoglobin concentration, mean corpuscular hemoglobin, red distribution width, platelet count, mean platelet volume, platelet hematocrit, and white blood cell count (results not shown) according to Baric Rafaj et al. [43]. Uterine tissues (endometrium and myometrium) and blood samples from the heart were collected post mortem up to 15–20 min after shooting from wild red deer females in the Strzałowo Forest District (hunting season 2018/2019; hunting license: ZG7521-3/2018/2019; N = 8), and directly after slaughter on a farm in Rudzie (north eastern of Poland; N = 21). The first experimental group represented the 4th day of the estrous cycle (N = 7), and the second group represented the 13th day (N = 8); the specimens were obtained on the 17th and 23rd of September 2019 after pharmacological synchronization on the farm in Rudzie by applying a double controlled internal drug-release (CIDR) insert (1.38 g of P4; Pfizer Animal Health, New York, NY, USA). Estrus and ovulation were induced by using a 12-day regimen of CIDR devices, which were replaced after 7 days to maintain the concentration of P4 until the end of the treatment period. On day 12, 200 IU of human chorionic gonadotropin (hCG; Folligon, Intervet, International B.V., Boxmeer, Holland) was injected intramuscularly. The estrus was observed 54–56 h after the removal of the second CIDR insert. The day of the estrous cycle was estimated by macroscopically observing the ovaries and uterus, and established by determining E2 and P4 levels in the blood plasma using an enzyme-linked immunosorbent assay (ELISA). The third group represented the anestrus stage (N = 6); non-pregnant females were obtained from the farm in Rudzie on the 10th of May (2019). The last experimental group, representing pregnant hinds (N = 8), was collected between the 2nd and 4th of January (2019) from wild females in Strzałowo. Pregnancy was confirmed by the presence of an embryo and additionally by the determination of *pregnancy*-associated *glycoprotein* concentration using EIA according to Kotlarczyk et al. [42]. The uterine tissue was placed on ice and transported in liquid nitrogen to the laboratory, where it was stored at −80 °C for further analysis. Blood samples were collected by jugular venipuncture in EDTA-loaded vacuum tubes. The samples were held on ice until centrifugation at 3000× *g* at 4 °C for 10 min, after which the plasma was transported to the laboratory within 30 min and stored at −20 °C until analyses (hematology, ELISA), whereas flow cytometry analysis was performed after the direct isolation of lymphocytes from the blood after the samples arrived at the laboratory.

### 4.2. Experimental Procedure

#### 4.2.1. Determination of T and B Lymphocytes’ Immunophenotypes in the Blood

Peripheral blood mononuclear cells (PBMCs) were isolated by density gradient centrifugation [44], washed, and frozen in freezing medium containing fetal bovine serum (FBS; Gibco, Life Technologies, Bleiswijk, The Netherlands) and 10% DMSO (Sigma-Aldrich, St. Louis, MO, USA) at −80 °C. Then, cell culture was performed in RPMI 1640 Medium with GlutaMAX™ (Gibco, Life Technologies, Bleiswijk, The Netherlands) containing 10% FBS, penicillin (100 IU/mL), streptomycin (100 μg/mL), nonessential amino acids (1%), MEM vitamins (100 μM), sodium pyruvate (1 mM), and amphotericin B (1 μg/mL) (Gibco™, Life Technologies, Bleiswijk, The Netherlands). A total of 4 × 10^6^ PBMCs were activated with a cocktail of phorbol 12-myristate 13-acetate (PMA) and ionomycin (eBioscience™, Invitrogen, Carlsbad, CA, USA; 5 μg/mL). The cells were incubated at 37 °C with 5% CO_2_ for 6 h.

Monoclonal antibodies with adequate fluorochromes were tested for labelling the CD4 (CVS4, BioRad, Hercules, CA, USA), CD8 (CVS21, BioRad, Hercules, CA, USA), CD21 (CC21, BioRad, Hercules, CA, USA), CD25 (IL-A111, BioRad, Hercules, CA, USA), and FoxP3 (FJK-16s, Life Technologies, Bleiswijk, The Netherlands; Table 1). The appropriate amount and concentration of each antibody was determined empirically in order to obtain optimal labelling results. The controls included unlabeled cells, and FMO (fluorescence minus one) and “switch-off” approach (SWOFF) controls were performed if necessary. For the analysis of lymphocytes, only non-adherent cells were collected. The cells were incubated with antibodies for 20 min at 4 °C in eBioscience™ Flow Cytometry Staining Buffer (Life Technologies, Bleiswijk, The Netherlands) in the dark. After that, the cells were washed twice with 2% BSA, resuspended in 200 μL of flow cytometry staining buffer, and immediately introduced into the cytometer. For FoxP_3_ staining, the eBioscience™ Foxp3/Transcription Factor Staining Buffer Set (Life Technologies, Bleiswijk, The Netherlands) was used according to the manufacturer’s protocol. The analysis was performed by setting the gate on single cells on the FSC-area (FSC-A) vs. FSC-high (FSC-H) dot plot. Based on the FSC and SSC dot plots, the lymphocytes were gated (Figure 5). The flow cytometric analysis was performed using a FACSCanto II flow cytometer and the Kaluza Flow Cytometry Analysis Software (Beckman Coulter, Brea, CA, USA).

#### 4.2.2. IgG, IgM, Haptoglobin, 6-keto-PGF1α, and cAMP Concentration in Plasma

IgG, haptoglobin, 6-keto-PGF1α, and cAMP concentration in the red deer females’ blood plasma was determined using ELISA.

#### 4.2.3. Arachidonic Acid Metabolites’ Abundances in Red Deer Uterus

PTGS2, PGES, PGFS, PGIS, 5-LO, LTA4H, and LTC4S mRNA and protein expression on selected days of the estrous cycle, anestrus, and pregnancy were determined in the uterine endometrial and myometrial tissue using real-time RT-PCR and Western blotting.

### 4.3. Determinations

#### 4.3.1. Total RNA Isolation and Reverse Transcription

Uterine tissue was homogenized in liquid nitrogen, then the total RNA was isolated with TRI Reagent (Sigma-Aldrich, Darmstadt, Germany, T9424) according to the manufacturer’s instructions. After extraction, the purity and RNA concentration was assessed using a NanoDrop 1000 spectrophotometer (Thermo Fisher Scientific, Wilmington, DE). The 260/280 nm wavelength ratios for all the samples were close to 2.0, and the 260/230 nm ratios ranged between 1.8 and 2.2. RNA. After the spectrophotometric measurement, 1 μg was reverse-transcribed using the High Capacity cDNA Reverse Transcription Kit (Applied Biosystems, Cheshire, UK, 4368813), which contained a MultiScribe™ reverse transcriptase with random primers, an RNase Inhibitor, MgCl_2_, a dNTP mixture, and nuclease-free H_2_O. The samples were incubated at 25 °C for 10 min, followed by 37 °C for 2 h. Finally, to inactivate the reverse transcriptase, the temperature was increased to 85 °C for 5 min. The obtained cDNA was kept at −20 °C until further analysis.

#### 4.3.2. Real-Time PCR

The mRNA expression of 5-LO, LTA4H, LTC4S, PTGS2, PGES, PGFS, and PGIS enzymes in the endometrium and myometrium was analyzed through RT-PCR using the Applied Biosystems Real-Time 7900 system (Applied Biosystems, Cheshire, UK), with the SensiFAST SYBR Hi-ROX Kit (Bioline Reagents, London, UK, BIO92002) according to the manufacturer’s instructions. The final PCR mix (10 µL) contained 3 µL of reverse-transcribed cDNA (15 ng), 5 µL of SensiFAST SYBR Hi-ROX Mix (SYBR Green and 3 mM MgCl_2_), and 0.2 µL of forward and reverse primers (at 0.5 μM concentration). Each run was performed in duplicate, and the average was considered as a single sample. The results were normalized according to the best reference gene, glyceraldehyde 3-phosphate dehydrogenase (GAPDH), chosen in preference over two other genes (β-actin and 18S ribosomal RNA), using the NormFinder software (Aarhus University, Denmark). The primer sequences are presented in Table 2. To evaluate the efficiency, standard curves based on serial dilutions of the cDNA were plotted, and the best cDNA concentration was chosen for further analysis. The first stage of the reaction was the initial denaturation of the strand and activation of the polymerase (95 °C for 2 min). The next stage consisted of 45 cycles of successive reactions: denaturation (95 °C for 5 s), primer annealing, and the elongation of the PCR products (6 °C for 20 s). To ensure the reaction’s specificity, the melting curves of the PCR products were analyzed after the amplification was completed. The data obtained were analyzed using the Miner program. Control reactions lacking the template or primers were performed to confirm that the products were free of primer dimers and contamination with genomic DNA, respectively.

#### 4.3.3. Total Protein Isolation

The uterine tissue (30 mg; endometrium and myometrium separate) was homogenized on ice in RIPA lysis buffer (5 mM EDTA, 150 mM NaCl, 50 mM TRIS, 0.1% SDS, 1% Triton X-100, 0.5% sodium deoxycholate, and a protease inhibitor from Sigma-Aldrich, S8830, pH 7.4). The obtained lysate was centrifuged at 10,000× *g* for 20 min at 4 °C, and the supernatant was transferred to a fresh tube and sonicated. The protein concentration was estimated according to Bradford’s method. The lysate was stored at −80 °C until further analysis.

#### 4.3.4. Western Blot Analysis

The expression of 5-LO, LTA4H, LTC4S, PTGS2, PGES, and PGIS proteins in the uterine tissues were determined by Western blotting, respectively. For each sample, 30 µg of total protein was mixed with 5 µL of SDS gel-loading buffer, heated at 95 °C for 5 min, and separated in a 10% SDS-PAGE gel. Afterward, the proteins were transferred to 0.2 μm nitrocellulose membranes in transfer buffer by electroblotting for 1.5 h. The membranes were blocked in a 5% solution of skimmed milk with 1x TBS-T for 1.5 h at room temperature (RT), and then incubated overnight at 4 °C with specific primary antibodies for 5-LO (Cayman, 160402, 1:1000), LTA4H (Cayman, 160250, 1:1000), LTC4S (Biorbyt, orb580627, 1:1000), PTGS2 (Cayman, 160107, 1:1000), PGES (Cayman, 160140, 1:1000), PGFS (Cayman, 10007940, 1:1000), and PGIS (Cayman, 160640, 1:1000). As a reference, the protein β-actin (ACTB, Sigma-Aldrich, A2228, 1:1000) was used. After that, the membranes were washed three times for 10 min in a 1x TBS-T solution and incubated with secondary polyclonal anti-rabbit antibodies for 5-LO, LTA4H, LTC4S, PTGS2, PGES, PGFS, and PGIS enzymes (Sigma-Aldrich, A3687, 1:10,000) and anti-mouse for ACTB (Sigma-Aldrich, A3562, 1:10,000) for 1.5 h at RT. The protein bands were visualized using AP buffer and NBT-BCIP solution (Sigma-Aldrich, 72091). Western blots were quantitated using the Quantity One 1-D Analysis Software (Bio-Rad, Hercules, CA, USA). 

#### 4.3.5. ELISA

The concentration of IgG, non-acetylated cAMP, haptoglobin, and 6-keto prostaglandin F1α were determined in the plasma using commercially available kits after validation. Each run was performed in triplicate, and the average was considered as a single sample.

The IgG assay (Biorbyt, United Kingdom, Cambridge, orb777055) range was estimated to be 1.57 to 100 μg/mL, and the sensitivity was 0.55 μg/mL. The absorbance was read at a 450 nm wavelength.

The non-acetylated cAMP assay (Cell Biolabs, Inc., San Diego, CA, USA, STA-500-5) range was estimated to be 100 pM to 100 µM, and the sensitivity was 1 pM/mL. The absorbance was read at a 450 nm wavelength.

The haptoglobin assay (Biorbyt, orb777055) range was estimated from 0.781 to 50 μg/mL, and a sensitivity of 0.469 μg/mL. absorbance was read at 450 nm length.

The 6-keto prostaglandin F1α assay (Cayman Chemical, Ann Arbor, MI, USA, 515211) range was estimated to be 1.6 to 1000 pg/mL, and the sensitivity was 6 pg/mL. The absorbance was read at a 420 nm wavelength.

#### 4.3.6. Statistical Analysis

GraphPad PRISM (Version 8.3.0, San Diego, CA, USA) was used for data analysis. The percentages of positive cells—CD21, CD4 and CD8, CD25 and CD4, FoxP3, and CD4—gated from the total lymphocytes and the relationship between the mRNA and protein expression of the endometrium and that of the myometrium, and between that of each experimental group throughout the endometrium and myometrium were determined using two-way analysis of variance (ANOVA) followed by Tukey’s test. One-way ANOVA was used for the analysis of IgG, cAMP, haptoglobin, and 6-keto-PGF1α concentration, followed by the Kruskal–Wallis test. All the numerical data are expressed as the arithmetic mean ± standard error of mean (SEM). Statistical significance was considered as *p* < 0.05.

## 5. Conclusions

Hinds, as seasonally reproducing ruminants, exhibit changes in the uterine regulation of AA metabolite action dependent on the reproductive status compared with ruminants without clearly marked anestrus. Immunological tolerance is changed in pregnancy, which is reflected by the higher percentage of CD4+CD25+ and CD4+FoxP3+ Tregs subpopulations in the blood of pregnant females in comparison to those in the estrus and anestrus group. Nevertheless, in future studies, it is necessary to evaluate the local immune response in the uterus to complement these results for the blood as well as the profile of cytokine production. We showed that, in the uterus, PTGS2 expression in hinds is constant during particular reproductive stages, but PGFS, PGES, and PGIS expression is mostly upregulated in pregnancy. The expression of the 5-LO pathway was variable throughout the examined groups, but its upregulation at the beginning of the cycle indicates participation in the regulation of uterine blood flow. The levels of selected immune parameters in the blood—IgG, cAMP, haptoglobin, and 6-keto-PGF1α—turned out to be good candidates for markers of condition dependent on the reproductive season in red deer females. The red deer were found to be a convenient experimental model for the recognition of immunocompetence, immune diagnostic, and reproductive function for other females of seasonally reproducing ruminants. What is more, it has been shown that physiological correlations between those processes are dependent on the reproductive stages, and fluctuations in the measured blood parameters do not originate from pathology. The results contribute to a better understanding of the mechanisms underlying seasonal reproduction in ruminants.

## Figures and Tables

**Figure 1 ijms-24-04771-f001:**
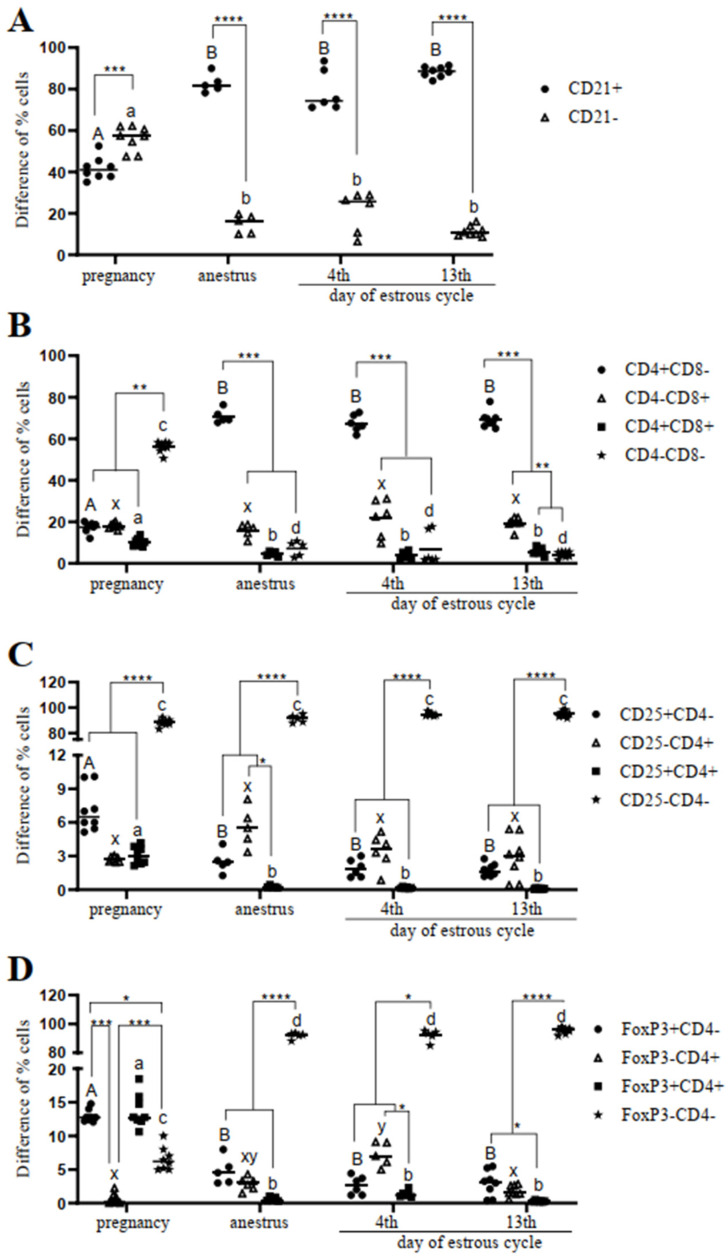
Percentages of positive cells: CD21 (**A**), CD4 and CD8 (**B**), CD25 and CD4 (**C**), and FoxP3 and CD4 (**D**) gated from total lymphocytes. Each dot represents one individual hind and means ± SEM (standard error of the mean) are presented. Statistical differences were analyzed with two-way analysis (ANOVA) of variance followed by Tukey’s post hoc test. The significance levels between gates are: * *p* < 0.05; ** *p* < 0.01, *** *p* < 0.001, and **** *p* < 0.0001 inside one experimental group. Different letters indicate statistical differences (*p* < 0.05) between the experimental reproductive stages.

**Figure 2 ijms-24-04771-f002:**
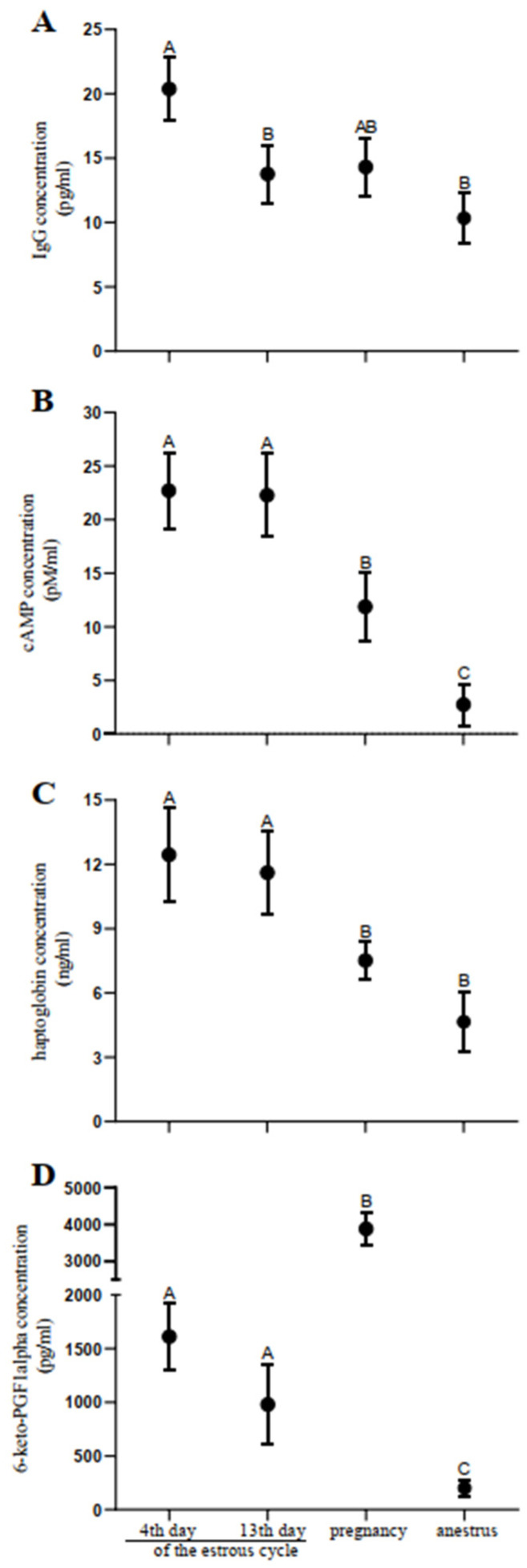
Plasma concentration of IgG (**A**), haptoglobin (**B**), cAMP (**C**), and 6-keto-PGF1α (**D**) in the blood samples collected on 4th and 13th day of the estrous cycle, and during pregnancy and anestrus. Plasma concentrations were estimated using ELISA kits, according to the manufacturer’s instructions. Statistical differences were analyzed with one-way ANOVA, followed by the Kruskal–Wallis test, using GraphPad PRISM (Version 8.3.0). The lowest statistical significance indicated by different letters (A–C) was *p* < 0.05.

**Figure 3 ijms-24-04771-f003:**
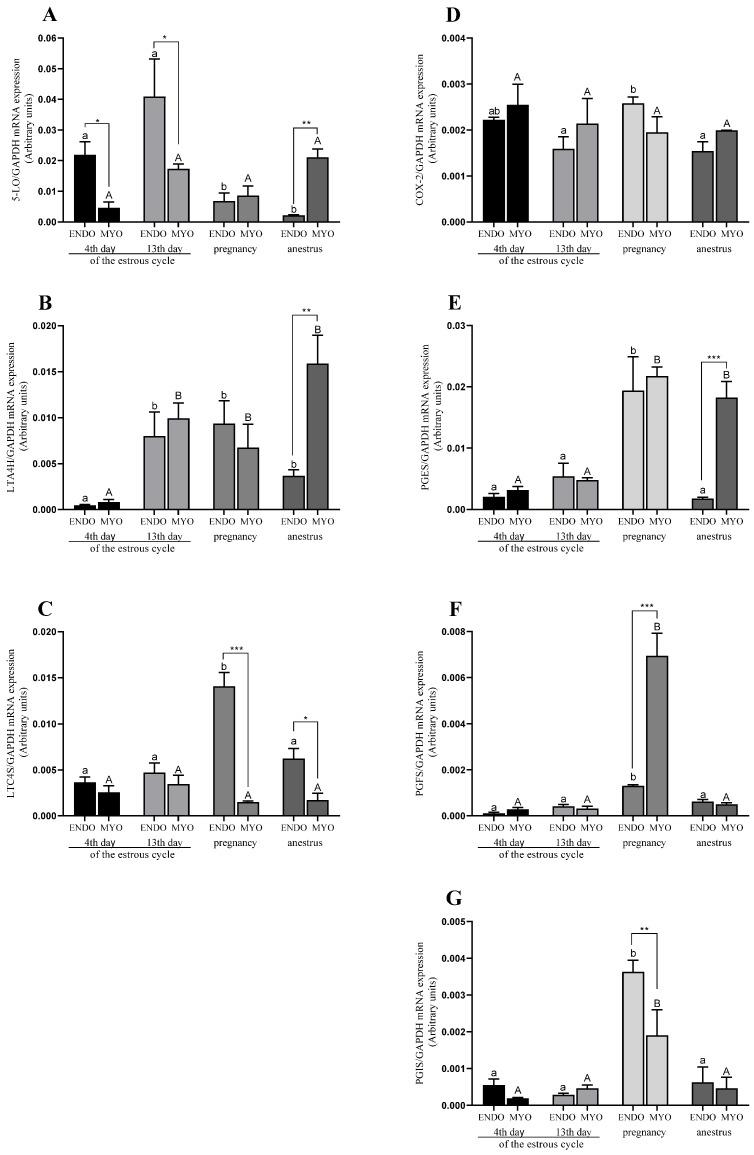
mRNA expression of 5-LO (**A**), LTA4H (**B**), LTC4S (**C**), PTGS2 (**D**), PGES (**E**), PGFS (**F**), and PGIS (**G**) in uterine tissues (endometrium and myometrium) on 4th and 13th day of the estrous cycle, and in pregnancy and anestrus phase. Data were normalized against GAPDH for mRNA expression using GraphPad PRISM (Version 8.3.0). Each bar represents one experimental group with SEM. Statistical differences were analyzed withy two-way analysis (ANOVA) of variance followed by Tukey’s post hoc test. The lowest statistical significance was *p* < 0.05. Asterisks indicate statistical differences between endometrium and myometrium (* *p* < 0.05; ** *p* < 0.01; *** *p* < 0.001). Different letters indicate statistical differences (*p* < 0.05) between the experimental reproductive stages throughout endometrium (a, b) and myometrium (A, B), respectively.

**Figure 4 ijms-24-04771-f004:**
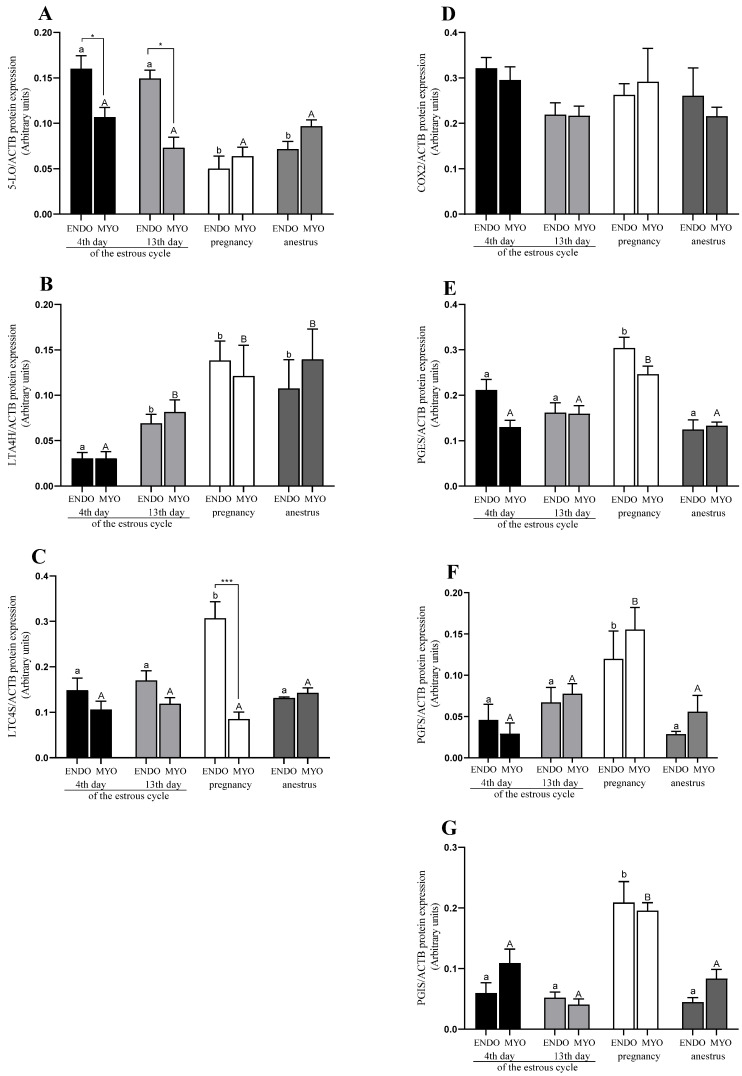
Protein expression of 5-LO (**A**), LTA4H (**B**), LTC4S (**C**), PTGS2 (**D**), PGES (**E**), PGFS (**F**), and PGIS (**G**) in uterine tissues (endometrium and myometrium) on 4th and 13th day of the estrous cycle, and in pregnancy and anestrus phase. Data were normalized against ACTB for protein expression using GraphPad PRISM (Version 8.3.0). Each bar represents one experimental group with SEM. Statistical differences were analyzed with two-way analysis (ANOVA) of variance followed by Tukey’s post hoc test. The lowest statistical significance was *p* < 0.05. Asterisks indicate statistical differences between endometrium and myometrium (* *p* < 0.1; *** *p* < 0.001). Different letters indicate statistical differences (*p* < 0.05) between the experimental reproductive stages throughout endometrium (a, b) and myometrium (A, B), respectively.

**Figure 5 ijms-24-04771-f005:**
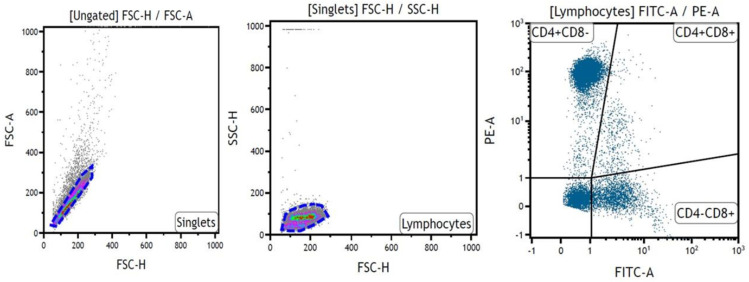
Example of lymphocyte gating strategy for CD4 and CD8 labelling. Doublets were removed from the analysis by setting the gate on single cells on the FSC-area (FSC-A) vs. FSC-high (FSC-H) dot plot. Next, the lymphocytes were gated based on FSC and SSC dot plots. Then, analysis of CD4+ and CD8+cells was conducted.

**Table 1 ijms-24-04771-t001:** List of monoclonal antibodies used for labeling lymphocytes for flow cytometry.

Gene Name	Primers Sequence (5′-3′)	Amplicon Length (bp)	EMBL
GAPDH	F: CACCCTCAAGATTGTCAGCA	103	BC102589
	R: GGTCATAAGTCCCTCCACGA		
ACTB	F: CCAAGGCCAACCGTGAGAAAAT	256	K00622
	R: CCACATTCCGTGAGGATCTTCA		
RN18S1	F: AAGTCTTTGGGTTCCGGG	365	AF176811
	R: GGACATCTAAGGGCATCACA		
5-LO	F: CACAGACGCAAAGAACTGGA	240	AJ306424
	R: CAGATTGTCTGGCAGCTTCA		
LTA4H	F: CCCTAAAGAACTGGTGGCACT	240	NM00103428
	R: GACTTTTCCACCTGCTCTTTC		
LTC4S	F: CCTGCTGCAAGCCTACTTCT	137	NM001046098
	R: GTTCACTTGGGCTCGGTAGA		
PTGS2(PTGS2)	F: TTGATTGAGAGTCCGCCAAC	158	NM174445
	R: GCAGTCATCAGGCACAGGAG		
PGES	F: CCCAAATTTGCACGTTCTCC	158	NM174443
	R: CCTGCAGTTTCAAGTGGGAC		
PGFS	F:AGTCGGAGGAGCAAAACAGA	169	S54973
	R:AATTTGGTGACCTCCACAGC		
PGIS	F: TCCTTTTGGGAGCAGAGCAG	103	L34208
	R: CTGAGGCTCTCACTCAGCAC		
	R: GTAGGCGTGGTTGATGGC		

**Table 2 ijms-24-04771-t002:** Oligonucleotide sequences used for real-time PCR. GAPDH—Glyceraldehyde 3-phosphate dehydrogenase, RN18S1—18S ribosomal RNA, ACTB—β-actin, 5-LO—5-Lipoxygenase, LTA4H—Leukotriene A4 hydrolase, LTC4S—Leukotriene C4 Synthase, PTGS2—prostaglandin endoperoxide synthase 2, PGES—Prostaglandin E2 synthase, PGFS—Prostaglandin F2alpha synthase, PGIS—Prostaglandin I2 synthase.

Antibody Name	Producer
CD4	CVS4; BioRad, Hercules, CA, USA
CD8	CVS21; BioRad, Hercules, CA,USA
CD21	CC21; BioRad, Hercules, CA, USA
CD25	IL-A111, BioRad, Hercules, CA, USA
FoxP3	FJK-16s, Life Technologies, Bleiswijk, The Netherlands

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
