# Peer review of "How Is Arachidonic Acid Metabolism in the Uterus Connected with the Immune Status of Red Deer Females (Cervus elaphus L.) in Different Reproductive Stages?"

_ijms, 2023, doi:10.3390/ijms24054771_

Round 1

Reviewer 1 Report

The paper refers to an interesting and important theme related to arachidonic acid metabolism in the uterus and immune status of red deer females during different reproductive stages. Results can contribute to a better understanding of mechanisms underlying a seasonal reproduction in ruminants.

 Specific remarks:

 L48-49 please delete the sentence

L96 please explain the abbreviation 5-LO when used for the first time

L105-106 I would suggest to use "reproductive stages" instead of "groups"

L149 please use "metabolite abundances" instead of "action"

L153, 213 the comment as in line 106

L337-338 please delete "(Komisja ds. Dobrostanu.......w Olsztynie)"

L407 add ", respectively." after Western blotting

L415 please delete "RNA."

L497-499 please rewrite the first sentence in CONCLUSION to make it more clear.

Author Response

Dear Reviewer,

On behalf of all the authors of the manuscript, thank you for your substantive review.
We have incorporated all the suggestions indicated into the manuscript.
Also, we have simplified the first sentence in Conslusion, which in the revised version reads: "Hinds as seasonally reproducing ruminants exhibit changes in the uterine regulation of AA metabolite action dependently on the reproductive status compared with ruminants without clearly marked anestrus."

Reviewer 2 Report

The manuscript is original, well organized and the results are discussed properly. However, some issues need to be clarified:

Comment 1. Please add the relevance of the study to the field in the abstract and conclusions.

Comment 2. The figures are illegible, not very clear, they should be corrected.

Author Response

Dear Reviewer,
On behalf of the authors, I kindly thank you for your review of the manuscript.

Answer for comments:

Comment 1. Please add the relevance of the study to the field in the abstract and conclusions.

We added to abstract the following sentence: "The results help expand our knowledge of the mechanisms underlying seasonal reproduction in ruminants."

We also added to Conclusion the following sentence: "Results contributed to a better understanding of mechanisms underlying a seasonal reproduction in ruminants."

Comment 2. The figures are illegible, not very clear, they should be corrected.

We discussed many times between the co-authors about the shape, type and presentation of the results, especially since there are relatively many of them (4 research groups and two types of uterine tissues), and we attempted to present the results in many ways.
However, I would like to refer to our previous article on steroid synthesis (Scientific Reports, DOI: 10.1038/s41598-021-99601-5), which was based on the same research groups and studied steroidogenesis in the same tissues. In that manuscript, we presented the results in a similar way as we do in this paper. We would like to keep the same pattern of presenting the results.
After Your suggestion, we have tried again to present the results in the form of tables, but, in our opinion, the results then lose the illustration of the magnitude and multiplicity of changes, especially when they are two- or threefold changes between the study groups. 
We kindly ask you to accept the results in their current form. Analyzing and delving into the first graph is certainly a bit tedious, but it will allow to understand the subsequent graphs easily because we have kept the same presentation model in all graphs throughtout the results (graph type, color, layout).